# Antioxidant Activity and Acute Oral Toxicity of Soursop (*Annona muricata* L.) Leaf and Its Effect on the Oxidative Stability of Mexican Hairless Pork Patties

**DOI:** 10.3390/foods14183212

**Published:** 2025-09-16

**Authors:** Pedro de Jesús Deniz-González, Fernando Grageola-Núñez, Pedro Ulises Bautista-Rosales, Armida Sánchez-Escalante, Gabriela María Ávila-Villarreal, Mario Estévez, Javier Germán Rodríguez-Carpena

**Affiliations:** 1Programa de Doctorado en Ciencias Biológico Agropecuarias, Universidad Autónoma de Nayarit, Km 9 Carretera Tepic-Compostela, Xalisco 63180, Nayarit, Mexico; pedro.deniz@uan.edu.mx; 2Unidad Académica de Medicina Veterinaria y Zootecnia, Universidad Autónoma de Nayarit, Km 3.5 Carretera Compostela—Chapalilla, Compostela 63700, Nayarit, Mexico; fgrageola@uan.edu.mx; 3Unidad de Tecnología de Alimentos, Secretaría de Investigación y Posgrado, Universidad Autónoma de Nayarit, Ciudad de la Cultura S/N, Colonia Centro, Tepic 63000, Nayarit, Mexico; ubautista@uan.edu.mx; 4Coordinación de Tecnología de Alimentos de Origen Animal (CTAOA), SECIHTI—Centro de Investigación en Alimentación y Desarrollo (CIAD), Carretera Gustavo Enrique Astiazarán Rosas 46, Hermosillo 83304, Sonora, Mexico; armida-sanchez@ciad.mx; 5Unidad Académica de Ciencias Químico Biológicas y Farmacéuticas, Universidad Autónoma de Nayarit, Tepic 63000, Nayarit, Mexico; gaby.avila@uan.edu.mx; 6Centro Nayarita de Innovación y Transferencia de Tecnología A. C. “Unidad Especializada en I+D+i en Calidad de Alimentos y Productos Naturales”, Universidad Autónoma de Nayarit, Tepic 63000, Nayarit, Mexico; 7Instituto de Investigación IPROCAR, Universidad de Extremadura, 10003 Cáceres, Spain; mariovet@unex.es

**Keywords:** soursop leaves extract, acute oral toxicity, meat color, oxidative stability of meat, Mexican hairless pig

## Abstract

The oxidation of meat and meat products can be delayed or mitigated through the use of natural antioxidants. Soursop leaf extracts have potential as a natural additive in meat products, offering a rich source of antioxidants. However, the impact and safety of incorporating soursop leaf extracts on the oxidative stability of meat products are not yet well understood. This study evaluated the antioxidant activity and acute oral toxicity of hydroalcoholic extracts from soursop leaves, as well as their effects on color and the oxidative stability of lipids and proteins in chilled Mexican Hairless pork patties. The results suggest that hydroalcoholic soursop leaf extracts may serve as a safe source of bioactive compounds with antioxidant properties, suitable for use as an additive in meat and meat products to reduce color loss and lipid oxidation, with a lesser effect on protein oxidation.

## 1. Introduction

Meat is an important food for human nutrition in most parts of the world, providing high-quality proteins, vitamins, and minerals with great bioavailability, in addition to lipids such as triglycerides and phospholipids [1,2,3]. However, despite its nutritional benefits, meat is a highly perishable food due to its complex chemical composition and intrinsic factors related to the animal. The main mechanisms of meat spoilage are driven by microbial activity, enzymatic processes, and the oxidation of lipids and meat proteins [4].

Oxidative processes have varying impacts on meat and meat products; a moderate level of oxidation can have positive effects on the development of typical flavors in some processed meat products, such as cured Iberian ham, cecina, and cured loin [5]. In contrast, a high level of oxidation reduces shelf life, leads to the deterioration of nutritional value, and compromises sensory qualities, such as color loss, texture alterations, and the appearance of unpleasant odors and flavors. Additionally, it promotes the formation of compounds with toxic effects on human health [2,4,6].

The oxidation of meat and meat products can be delayed or reduced through the use of antioxidant compounds. For many years, synthetic antioxidants such as butylated hydroxyanisole (BHA), butylated hydroxytoluene (BHT), and tertiary butylhydroquinone (TBHQ) were used as food additives to mitigate the negative effects of oxidation. However, these compounds have been associated with adverse health effects, including toxicological and carcinogenic risks [7,8]. Consequently, and in line with the growing interest in foods that provide benefits beyond mere nutrition, the demand for natural antioxidants that are cost-effective, safe for human health, and effective at low concentrations has recently increased in the meat industry [3,7,9].

Natural antioxidants are generally metabolic products and enzymes from the cells of living organisms that function to maintain the redox balance [9,10,11]. Natural antioxidants can be of animal origin, primarily from herbivorous species [11], and bacteria and fungi have also been reported as sources of antioxidant compounds [12,13]. However, the plant kingdom is the primary source of natural antioxidants used as additives in meat and meat products, as antioxidant compounds are extracted from multiple sources, such as fruits, vegetables, aromatic herbs, spices, tree leaves, seeds, and agricultural and industrial by-products [9,10,14]. In addition to providing antioxidant benefits, plant matrices and their extracts contain nutrients or bioactive compounds that are rarely present in meat and meat products, thereby enhancing their nutritional and functional quality [3].

The soursop (*Annona muricata* L.) is a fruit tree native to the Americas and belongs to the Annonaceae family. Currently, Mexico is the world’s leading producer of soursop. However, its distribution extends to regions in West Africa, Southeast Asia, and tropical areas of Central and South America [15,16,17]. Soursop fruit is primarily consumed fresh when it reaches maturity. The pulp is processed into products such as juices, ice creams, yogurts, and more [18]. The ground seeds of soursop fruit are known for their therapeutic effects against internal and external parasites [19]. Regarding the soursop tree, its bark is used in traditional medicine to treat inflammation, hypertension, parasitic infections, and hypoglycemia. The root is utilized in the production of biopesticides and bioinsecticides. The leaves are used in traditional medicine for therapeutic purposes, such as pain management and the treatment of hypertension, insomnia, cancer, asthma, colds, and malaria [20].

Although all parts of the soursop fruit and tree are used, the leaves are receiving particular attention regarding antioxidant activity because they contain bioactive compounds, including phenolic compounds such as flavonoids, phenolic acids, and gallotannins, which contribute to a potent antioxidant effect [16,20]. Additionally, in soursop cultivation, large quantities of branches and leaves are discarded annually, which could be utilized since pruning is an essential practice for optimizing fruit harvest [21].

The use of soursop leaf extracts as a natural additive in meat products is not currently documented. As a result, the impact of using soursop leaf extracts on the oxidative stability of meat products is unknown. This study evaluated the antioxidant activity and acute oral toxicity of hydroalcoholic extracts of soursop leaves, as well as their effect on color and oxidative stability when used as a natural additive in chilled Mexican Hairless pork patties.

## 2. Materials and Methods

### 2.1. Chemicals and Reagents

All reagents used were of analytical grade. Folin–Ciocalteu phenol reagent, gallic acid, 1,1-diphenyl-2-picrylhydrazyl (DPPH•), 2,2′-Azino-bis(3-ethylbenzothiazoline-6-sulfonic acid) diammonium salt (ABTS), (S)-6-Methoxy-2,5,7,8-tetramethylchromane-2-carboxylic acid (Trolox), 1,1,3,3-Tetraethoxypropane (TEP) butylated hydroxytoluene (BHT), 2,4-dinitrophenylhydrazine (DNPH) and Guanidine hydrochloride were purchased from Sigma Aldrich Chemicals (St. Louis, MO, USA). Sodium carbonate, ethanol, methanol, potassium persulfate, 2-thiobarbituric acid (TBA), perchloric acid, ethyl acetate, sodium phosphate dibasic 7-hydrate crystal, sodium phosphate monobasic monohydrate, Trichloroacetic acid (TCA) and hydrochloric acid (HCl) were obtained from JT Baker (Baker^®^, Phillipsburg, NJ, USA). The solvent for obtaining soursop leaf extracts was food grade ethanol (Alcohol Lord^®^, Dark Lord Brewery, Jalisco, México) purchased from a local supplier.

### 2.2. Vegetal Material

Four kilograms of soursop tree leaves were collected from a commercial soursop orchard located in Venustiano Carranza, Tepic, Nayarit, Mexico (21°30′ N, 104°54′ W, 920 m above sea level), in December 2020. The fresh leaves were cleaned with microfiber towels moistened with tap water and then stored in vacuum-sealed bags at −20 °C until use.

### 2.3. Antioxidant Activity Evaluation of Soursop Leaf Extracts

#### 2.3.1. Extraction of Fresh Leaf Extracts for Antioxidant Activity

Two different extracts were obtained from fresh soursop leaves, the first using ethanol (70:30 *v/v*) and the second using only water as the solvent. For extraction, three grams of finely chopped fresh, whole leaves were homogenized with 15 mL of the solvent using a Silent Crusher M (Heidolph^®^, Heildolph Instruments GmbH & Co., Schwabach, Germany). After homogenization, the tubes were centrifuged at 2500 rpm at 4 °C for three minutes, and the liquid fraction was collected in a ground-glass flask through Whatman No. 1 filter paper. The resulting residue was treated again with 15 mL of the same solvent, applying the same centrifugation and filtration conditions. The total filtrate was then evaporated using a rotary evaporator at a pressure of 175 mbar and a maximum temperature of 42 °C until the solvent completely evaporated. The aqueous resulting extract was redissolved in 25 mL of water and stored at 4 °C until further analysis.

#### 2.3.2. Determination of Total Phenolic Content and DPPH and ABTS

Total phenolic compounds were determined using the Folin–Ciocalteu method [22] with the diluted extract (1:20 *v/v*). Absorbance was measured at 765 nm, and the content was calculated based on a gallic acid standard curve (5 to 100 mg). The results were expressed as mg gallic acid equivalents per 100 g of fresh material.

Antioxidant activity was evaluated using the ABTS and DPPH• radical assays as reported by Rodríguez-Carpena et al. [22]. For the ABTS assay, absorbance was read at 734 nm, and for the DPPH assay, absorbance was read at 517 nm. For the ABTS assay, the calibration curve was constructed with various concentrations (0.25 to 2 mmol) using a Trolox ethanol standard solution and for the DPPH assay the calibration curve was constructed with various concentrations (0.25 to 2 mmol) using a Trolox methanol standard solution. Results were calculated and expressed as mmol Trolox equivalents per gram of fresh material.

### 2.4. Acute Oral Toxicity of Soursop Leaf Extract

The hydroalcoholic extract, which showed significantly higher phenolic compound content and better antioxidant activity, was selected for acute oral toxicity evaluation in Wistar rats.

#### 2.4.1. Extraction of Dried Leaf Extracts for Toxicity Testing

For the determination of acute oral toxicity, an extract was obtained from dried soursop leaves. Initially, one kilogram of fresh leaves was selected and dried in the shade at room temperature. Once the plant material was dry, a food processor (Oster^®^,Newell Brands de México LTD, D.F., Ciudad de México, México) was used to chop the leaves into small pieces. The dried plant material weighed 380 g. The chopped material was subjected to maceration extraction with a hydroalcoholic solution (ethanol-water 85:15 *v/v*) for 72 h. The extract was filtered through Whatman No. 1 filter paper, and the solvent was removed using rotary distillation at a pressure of 175 mbar until dry extracts were obtained. The entire maceration process was carried out in quintuplicate.

#### 2.4.2. Animals and Housing Conditions

The protocol for determining acute oral toxicity was approved by the State Bioethics Commission of the State of Nayarit (Registration No.: CEBN/13/18), and animal handling was carried out according to the standards established in NOM-062-ZOO-1999 [23]. Twelve adult male Wistar rats with an average weight of 444 ± 44 g were housed in plastic cages. The animals were fed ad libitum with a regular diet of commercial pellets and had free access to water, except on the day of administration. The laboratory was free of specific pathogens, and the conditions were controlled at a temperature of 22 ± 3 °C, relative humidity of 45–70%, and a 12 h/12 h light/dark cycle. All animals had a one-week adaptation period to laboratory handling and conditions before the toxicity study began.

#### 2.4.3. Acute Oral Toxicity

The acute oral toxicity of soursop leaf extract was investigated using the acute Toxic Class Method, following the OECD 423 guideline methodology [24]. This is a stepwise procedure (2 to 4 steps) using three animals of the same sex per step for both the test substance and the vehicle.

In this study, an initial dose of 2000 mg/kg was used in the first step. Six rats were divided into 2 groups of 3 animals each. Group I (control group) was orally administered a vehicle (ethanol–water 85:15 *v/v*), and Group II was administered 2000 mg/kg of soursop leaf extract. In the second step, six rats were divided into 2 groups of 3 animals each. Group I (control group) received a vehicle (ethanol–water 85:15 *v/v*) (vehicle), and Group II received 2000 mg/kg of soursop leaf extract. The test substance was prepared in a hydroalcoholic solution (ethanol–water 85:15 *v/v*) and administered as a single dose by oral gavage after the animals had fasted overnight before the test.

After dosing, the animals were kept fasting for the subsequent 4 h, and signs and behaviors, including changes in skin, fur, eyes, mucous membranes, and respiratory tract, as well as tremors, convulsions, salivation, diarrhea, lethargy, sleep, coma, and death, were observed according to OECD guideline 423 [24]. Clinical signs of toxicity, recovery period duration, and mortality were observed after dosing during the first 4 h and then daily for a period of 14 days. The animals body weight was recorded daily.

### 2.5. Oxidative Stability of Mexican Hairless Pork Patties with the Inclusion of Soursop Leaf Extract

The hydroalcoholic extract, which yielded significantly higher amounts of phenolic compounds and demonstrated better antioxidant activity, was used as an additive in the preparation of Mexican Hairless pork patties.

#### 2.5.1. Manufacture and Handling of Mexican Hairless Pork Patties

In the present experiment, eight Mexican Hairless pigs were used as experimental units. The pigs were raised under an intensive production system and fed conventional diets that met the nutritional requirements established by the National Research Council for Swine [25]. Once the pigs reached 12 months of age, they were slaughtered following the procedure described in NOM-033-SAG/ZOO-2014 [26].

The meat was obtained from the *Longissimus dorsi* muscle and dorsal fat of the same animal (24 h post-mortem). For each experimental unit (Mexican Hairless pig), two patties treatments were prepared: a control group (without the addition of extract) and a test group, which included the addition of soursop leaf extract. The formulation of the patty mixture per kilogram included the following ingredients: 800 g of Mexican Hairless pork, 88.5 g of Mexican Hairless pork dorsal fat, 11.5 g of iodized salt, and 100 g of distilled water.

Using an 8 cm diameter metal ring as a mold, three circular patties of each experimental units (pig = 8) of 100 g of the mixture described were made for each treatment (n = 24 per treatment, divided into three time periods). The patties were placed on polystyrene trays and covered with transparent polyvinyl chloride film (SUNwrap^®^) (Thickness (Mic): 10~50. Tensile Strength MD/TD (kgf/mm^2^): 3/2. Tensile Elongation MD/TD (%): 300/400. Haze: 0~0.4. Adhesion: Good. AntiFrogging: 100 °C/10~13 min. (oxygen permeability: ~17 cm^3^/m^2^ day atm; moisture permeability: <5 g/m^2^ day). They were then stored under refrigeration at 4 °C, with continuous fluorescent lighting (1620 lx) during the 10 days of the experiment, simulating retail conditions. The patties from each treatment were evaluated after 0, 5, and 10 days of storage (n = 8 per period of time and treatment). Once the patties reached the sampling period, they were stored at −20 °C (Figure 1).

#### 2.5.2. Obtaining and Applying Fresh Leaf Extracts as an Additive for Meat Products

The amount of fresh soursop leaves used to obtain the extract for the test patties group was 20 g per kg of mixture, which represents the equivalent of 48,536 mg GAE per kg of mixture. The extract was prepared using ethanol (70:30 *v/v*) and following the same methodology described in Section 2.3.1. Once the aqueous residue from the solvent rotary evaporation was obtained, the extract was brought to volume with water to complete the 10% portion of water required in the formulation.

#### 2.5.3. Color Measurements

Color measurements on the surface of the patties on days 0, 5, and 10 of chilled storage were performed using a Minolta CR-400 colorimeter (Minolta Camera Corp., Meter Division, Ramsey, NJ, USA), which consists of a measurement head with a 10 mm measurement area and a data processor. Before each measurement session, the colorimeter was calibrated in the CIE color space system [27] using a white tile. The result of one measurement was the average of three continuous color evaluations on different points of the sample’s surface. Color measurements were conducted at room temperature (≈22 °C) with D65 illumination and a 0° observer angle.

Based on the obtained color measurements, the total color differences (ΔE) were also calculated for both, the control group patties and the test group patties with the addition of soursop leaf extract. The ΔE was calculated between the different days of chilled storage for the same treatment (ΔE0–5), (ΔE5–10), and (ΔE0–10). The total color differences (ΔE) were calculated as follows:


ΔE5-0= [(L5 − L0) + (a5 − a0) + (b5 − b0)2]1/2
(1)



ΔE10-5 = [(L10 − L5) + (a10 − a5) + (b10 − b5)2]1/2
(2)



ΔE10-0 = [(L10 − L0) + (a10 − a0) + (b10 − b0)2]1/2
(3)


#### 2.5.4. Determination of Thiobarbituric Acid Reactive Substances (TBARS)

The quantification of malondialdehyde (MDA) from Mexican Hairless pork patties after 0, 5, and 10 days of chilled storage was carried out using the TBARS technique, following the thiobarbituric acid method [28]. The standard curve consisted of eight points prepared using a TEP solution (0.23 g/L), and absorbance was measured at 532 nm. The results were expressed as mg of MDA/kg of sample.

#### 2.5.5. Determination of Total Protein Carbonyls (DNPH)

Protein oxidation in Mexican Hairless pork patties after 0, 5, and 10 days of chilled storage was quantified using the total carbonyl method with 2,4-dinitrophenylhydrazine (DNPH) [28]. Protein concentration was calculated based on absorbance at 280 nm using bovine serum albumin as the standard. The concentration of carbonyls was expressed as nanomoles of carbonyls per milligram of protein using an absorption coefficient of 21 nM^−1^ cm^−1^ at a wavelength of 370 nm for protein hydrazones.

### 2.6. Statistical Analysis

Three independent experimental trials were conducted and all analytical procedures were carried out in triplicate for each sample. The data are expressed as the mean for each treatment and the standard error of the mean. The normality of the data was determined using the Shapiro–Wilk test.

To evaluate significant differences between treatments in the characterization of aqueous and hydroalcoholic soursop leaf extracts, acute toxicity test, and patty model system analyses Student’s *t*-test for independent samples was used, considering a 95% confidence level.

Pearson correlation analyses were conducted between antioxidant activity and oxidative stability variables. Statistical analyses were performed using the SAS System for Windows version 9.0.

## 3. Results

### 3.1. Total Phenolic Compounds and Antioxidant Activity of Soursop Leaf Extracts

The quantification of total phenolic compounds and antioxidant activity in the aqueous and hydroalcoholic extracts of soursop leaves are presented in Table 1. The results indicate that the extract obtained using ethanol (70:30 *v/v*) contains a significantly higher concentration of total phenolic compounds and greater antioxidant activity in the ABTS and DPPH assays compared to the aqueous extract.

A Pearson correlation matrix between the concentration of phenolic compounds and antioxidant activity is shown in Table 2. The value of total phenolic compounds is positively correlated with the values of both antioxidant activity assays. However, the correlation with each assay differs in intensity. Total phenolic compounds show a weak correlation with antioxidant activity obtained through the ABTS assay, while there is a moderate correlation between total phenolic compounds and antioxidant activity as measured by the DPPH assay. The antioxidant activity assays between ABTS and DPPH also maintain a positive correlation with moderate intensity.

### 3.2. Acute Oral Toxicity of the Hydroalcoholic Soursop Leaf Extract

In the first stage, neither the control group (n = 3) nor the group of animals administered with the soursop leaf extract (2000 mg/kg) (n = 3) showed any apparent signs of toxicity or behavioral alterations. There were no deaths in this initial step of the study.

In the second stage of the study, the control group also showed no apparent signs of toxicity. Regarding the animals administered with the soursop leaf extract (2000 mg/kg), two of them showed no apparent signs of toxicity or any alteration in behavior. However, one individual exhibited spasmodic jumps 70 min after administration, which persisted for four hours, followed by complete recovery after 12 h. This same individual showed clinical signs of dyspnea on the 13th day post-administration, recovering one hour after the onset of clinical signs. There were no deaths in the second stage of the study.

Regarding the monitoring of body weight in the rats participating in the acute oral toxicity evaluation (Table 3), both experimental groups showed a similar weight increase compared to their initial weight. The control group experienced a 7.27% increase in body weight, while the group of animals administered with the soursop leaf extract exhibited an 8.53% gain over the 14-day study period. The initial weight, weight at 7 days, and weight recorded at 14 days post-administration showed no significant differences between treatments.

The acute oral toxicity evaluation method outlined in OECD Guideline 423 [24] provides information on the adverse effects of substances while allowing their categorization based on the framework of the Global Harmonized System. Based on the absence of deaths, the soursop leaf extract is classified under Category 5 of the Global Harmonized System for the classification of chemicals, with an LD50 of 5000 mg/kg body weight. Substances in this category are considered “safe for human consumption” due to their relatively low acute oral toxicity, though they may pose a risk to vulnerable populations under certain circumstances.

### 3.3. Soursop Leaf Extract Impact on the Color Stability of Mexican Hairless Pork Patties

The evolution of instrumental color of the pork patties during chilled storage is presented in Table 4. The addition of soursop leaf extract significantly influenced the L*, a*, b*, C*, hue, and total color difference (ΔE) coordinates.

The lightness was significantly lower in the patties with added soursop leaf extract throughout the experimental period (days 0, 5, and 10). Regarding the red coloration of the meat, significant differences were only observed on day 0, with the soursop leaf extract-added patties showing a lower a* value compared to the control group patties. In the case of the b* axis, the b* value was significantly higher in the soursop leaf extract-added patties throughout the experimental period (days 0, 5, and 10).

In terms of Chroma, significant differences were observed at 5 and 10 days of storage, with the soursop leaf extract-added patties showing a higher C* value compared to the control group patties. Regarding hue, significant differences were observed on days 0 and 5 of storage, with the soursop leaf extract-added patties showing a higher hue value compared to the control group patties.

The total color differences (ΔE) between the different storage days for both the control patties and the soursop leaf extract-added patties are shown in Figure 2.

During the period between day 0 and day 5 of storage, patties from both treatments had similar color differences. During the period between day 5 and day 10 of storage, the control group patties experienced more intense color deterioration compared to the soursop leaf extract-added patties. Over the entire experimental period, from day 0 to day 10 of storage, the control group patties also experienced more intense color deterioration compared to the soursop leaf extract-added patties.

### 3.4. Soursop Leaf Extract Impact on Lipid Oxidation in Mexican Hairless Pork Patties

The evaluation of lipid oxidation through the quantification of MDA in the control patties and the soursop leaf extract-added patties during chilled storage is presented in Figure 3. The soursop leaf extract-added patties exhibited a lower amount of TBARS throughout the experimental period (days 0, 5, and 10). The behavior of lipid oxidation over storage time was similar in both the soursop leaf extract-added patties and the control group, with significantly different MDA/kg concentrations observed on each sampling day, showing a marked increase as storage days progressed.

### 3.5. Soursop Leaf Extract Impact on Protein Oxidation in Mexican Hairless Pork Patties

The quantification of protein carbonyls in the control patties and the soursop leaf extract-added patties during chilled storage is presented in Figure 4. Significant differences between treatments were observed on day 0 and day 5 of storage. On day 0, the carbonyl content was higher in the soursop leaf extract-added patties. However, on day 5 of storage, the carbonyl content was higher in the control group patties. On day 10 of storage, no significant differences were observed between treatments.

When evaluating the behavior of protein oxidation over storage time, it differed between both treatments. In the control group patties, an increase in protein carbonylation was observed between day 0 and day 5 of storage; however, a slight decrease was observed on day 10 of storage. In the soursop leaf extract-added patties, there was a reduction in protein carbonylation between day 0 and day 5 of storage, but protein carbonylation slightly increased on day 10 of storage.

### 3.6. Pearson Correlations Between Instrumental Color, Lipid Oxidation, and Protein Oxidation in Mexican Hairless Pork Patties with Added Soursop Leaf Extract

The evaluation of the oxidative stability variables (color, TBARS content, and total carbonyl content) through Pearson correlations is shown in Table 5.

Lightness showed a moderate negative correlation with the a*, b*, and C* values but a moderate positive correlation with hue and TBARS content. The a* value had strong positive correlations with C* and negative correlations with h*, and it also had a moderate negative correlation with TBARS content. The b* axis showed moderate positive correlations with C* and negative correlations with TBARS content.

Regarding Chroma and hue, Chroma had a strong negative correlation with TBARS content. The hue value had a moderate positive correlation with TBARS content.

Protein carbonylation did not show a moderate or strong correlation with the color variables or TBARS content. It only had weak correlations, such as a positive correlation with TBARS content and a negative correlation with the b* axis.

## 4. Discussion

### 4.1. Antioxidant Evaluation of Soursop Leaf Extracts

It is well documented that the content of bioactive antioxidant compounds in an extract depends on various factors, such as the extraction method, time, temperature, matrix, and solvent used [16,20]. For the extraction of phenolic compounds from *Annona muricata* L. leaves, it has been reported that polar solvents are predominantly used. Among the use of polar solvents, there is evidence that aqueous solutions with some proportions of ethanol are more effective at extracting antioxidant compounds than water alone [20,29]. However, there are also studies that show the opposite; for example, an aqueous extract of soursop leaves made in Uganda contained a higher concentration of phenolic compounds than the ethanolic extracts of the same leaves [30]. In this study, it was found that the use of ethanol (70:30 *v/v*) in fresh soursop leaves resulted in a significantly higher extraction of total phenolic compounds compared to the use of water alone as the extraction solvent.

The higher concentration of phenolic compounds in the extract obtained using ethanol (70:30 *v/v*) is due to a synergistic effect when mixtures of water and some type of alcohol are used in the extraction, as water swells the plant cells while ethanol causes dehydration and collapse of the plant cells, leading to the rupture of the cell wall and the loss of osmotic balance [31]. However, considering the classification criteria suggested by Vasco et al. [32], where the concentration levels of phenolic compounds are divided into low (20–100 mg/g FW), intermediate (200–1000 mg/g FW), and high (greater than 1000 mg/g FW), both the hydroalcoholic extract and the aqueous extract obtained in this study can be categorized at an intermediate level. This confirms that soursop leaves can be an important source of phenolic compounds [16,20].

Considering the type of phenolic compounds that might be present in the extracts obtained in the present experiment, information has been reported on soursop leaf extracts collected from different regions of Nayarit, Mexico, obtained through aqueous or ethanolic solutions, where the main compounds are phenolic acids and flavonoids, although there is also the presence of lignans and an allylbenzene [16,33,34]. The following phenolic acids have been reported: shikimic acid, gallic acid, protocatechuic acid, neochlorogenic acid, 3,4-dihydroxyphenylacetic acid, 4-hydroxybenzoic acid, chlorogenic acid, 4-hydroxyphenylacetic acid, vanillic acid, vanillin, syringic acid, 3-hydroxybenzoic acid, 4-hydroxybenzaldehyde acid, homovanillic acid, 3-propionic acid, p-coumaric acid, trans-ferulic acid, ferulic acid, sinapic acid, and ellagic acid. The main flavonoids reported are: gallocatechin, epigallocatechin, catechin, epicatechin, quercetin rutinoside (rutin), kaempferol dihexoside, quercetin hexoside, iso-rhamnetin rhamnoside, kaempferol hexoside-rhamnoside, quercetin rhamnoside, quercetin, kaempferol, and naringenin. Additionally, the presence of lignans such as secoisolariciresinol and an allylbenzene like eugenol has been reported [16,33,34].

There is information on antioxidant activity indicating that the concentration of phenolic compounds and the antioxidant activity of plant-based extracts determined by DPPH and ABTS assays have a strong positive correlation [30,32,35]. In this study, according to the correlation intensity classification of Ratner [36], the correlations between the concentration of phenolic compounds and the antioxidant activity obtained by the ABTS and DPPH assays showed a positive trend but with different intensities, manifesting a weak correlation between the value of phenolic compounds and the value of the ABTS assay, and a moderate correlation between the amount of phenolic compounds and the value of the DPPH assay. This suggests that the phenolic compounds extracted scavenged a greater number of radicals in a hydrophobic environment, as the DPPH assay uses the DPPH· radical, which is dissolved in organic media and is better suited for hydrophobic antioxidant systems. On the other hand, the ABTS technique generates its own ABTS+ radical by chemical reaction, which is blue/green in color and applies to both lipophilic and hydrophilic antioxidant systems [37]. Moreover, the similarity in the principle of the ABTS and DPPH assays often results in a strong positive correlation between the results of both techniques in different matrices [30,32,35,37]. In this research, it was found that the results of the DPPH and ABTS tests had a moderate positive correlation and therefore similar behavior in both assays, showing greater antioxidant activity in the hydroalcoholic extract compared to the aqueous extract.

The bioactive compounds in soursop leaf extracts that contribute to antioxidant activity are mainly flavonoid-type phenolic compounds [29]. The group of flavonoids with proven antioxidant activity are apigenin-6-C-glucoside, argentinine, catechin, homoorientin, kaempferol, kaempferol 3-O-rutinoside, robinetin, quercetin, quercetin 3-O-glucoside, quercetin 3-O-neohesperidoside, quercetin 3-O-robinoside, quercetin 3-O-rutinoside, epicatechin, luteolin 37-di-O-glucoside, and quercetin 3-α-rhamnosyl, robinetin [20,33]. Also, caffeic acid has proven antioxidant action, but derivatives of hydroxycinnamic acid play a lesser role in antioxidant activity. Other non-phenolic bioactive compounds present in these extracts and attributed with antioxidant activity are vitamin A (carotenoids) and vitamin E (tocopherols) [29].

### 4.2. Acute Oral Toxicity of the Hydroalcoholic Extract of Soursop Leaves

Due to the increased traditional therapeutic use of soursop and the emergence of an atypical Parkinson’s disease in the French Antilles associated with the consumption of soursop, considerable information has emerged regarding the toxicity of soursop extracts, as well as bioactive compounds present in different parts of the tree and fruit [29,38].

Regarding aqueous extracts of soursop leaves, the oral toxicity level in murine models is low, showing no apparent signs of toxicity, with LD50 values greater than 5000 mg/kg and 800 mg/kg for an extract from leaves obtained in Cameroon, administered over 14 and 30 days, respectively. Additionally, the administration of an extract from leaves sourced from Thailand, with doses ranging from 100 mg/kg to 4000 mg/kg in mice over 7 days, has been reported. Signs of toxicity were observed at 4000 mg/kg, and an LD50 greater than 2500 mg/kg was noted [39].

In terms of the reported toxicity levels for ethanolic extracts of soursop leaves, in an experiment involving 50 mice, they were administered a dried soursop leaf extract obtained using 95% ethanol, and an LD50 of 1670 mg/kg was reported [40]. In another experiment where an ethanolic extract of soursop leaves from Brazil was also administered, deaths among the experimental units were reported, and an LD50 ≥ 1000 mg/kg was noted [40]. There are also studies suggesting a low level of toxicity in ethanolic extracts of soursop leaves. In an experiment conducted on mice, where the oral toxicity level in vivo of the ethanolic fraction of a dried soursop leaf extract obtained through maceration with a 98% ethanolic solution was evaluated, no significant changes were found in the plasma levels of hepatic injury markers (AST and ALT) and renal (creatinine), as well as in hematological profiles (red blood cells, white blood cells, and platelets) [15]. Additionally, in another experiment with mice, three types of soursop leaf extracts were administered: pure soursop leaf extract, soursop leaf extract enriched with flavonoids, and soursop leaf extract enriched with acetogenins. The concentration of the administered extracts was 2000 mg/kg. The experimental units administered with the pure extract and the extract enriched with flavonoids showed no apparent signs of toxicity or deaths. However, all the experimental units administered with the extract enriched with acetogenins died [39], suggesting the toxicity of these bioactive compounds. In fact, regarding the toxicity of isolated bioactive compounds from soursop, there is information indicating that a group of acetogenins (mainly annonacin) and alkaloids (mainly reticuline) present in the leaves and fruit of soursop act as neurotoxins that generate neurodegenerative disorders in murine models [29,38,40].

In general, there is a wide variation in the toxicity levels of soursop leaf extracts, with both high and low toxicity levels being observed. In our study, we found that the hydroalcoholic extract of soursop leaves had an LD50 of 5000 mg/kg b.w., which is considered safe for human consumption according to OECD guidelines. This low toxicity of our extract also aligns with reports of the absence of toxicity in *Artemia salina*, as observed in various extracts of soursop leaves collected in the state of Nayarit, México, in 2021. The extraction processes in this study included decoction, infusion, and extraction using an acetone solution (80:20 *v/v*) assisted by ultrasound [16]. Based on this, it can be considered that the reported toxicity of soursop leaf extracts is variable and likely depends on the environmental and geographical conditions of the plant, the type of animal model used, and the solvent and extraction method employed, as all these factors influence the type and quantity of bioactive compounds extracted [16,29,41].

### 4.3. Color Deterioration of Mexican Hairless Pork Patties

Meat color is one of the most important visual aspects, as it is usually the main factor influencing consumers’ purchasing decisions at the point of sale [42]. This is primarily because consumers associate the color of meat with its freshness, safety, and sensory quality [43,44,45].

The L* axis value is characterized by changes in response to achromatic stimuli [46,47]. In this study, it was observed that the addition of soursop leaf extract decreased the L* value of the patties throughout the experimental period compared to the L* value of the control patties. This trend in L* values was also observed in an experiment where the lightness of patties supplemented with different concentrations of red dragon fruit extract was compared with the control group [48]. The aforementioned phenomenon of our experimental samples could be attributed to the bioactive compounds and pigments present in the soursop leaf extract added to the patties, which have a higher light absorption capacity. This greater light absorption causes less light scattering by particles or structures located between the muscle fibers and the myofibrils near the surface. It has been reported that if the surface scattering is low, the material will appear darker on the L* axis [46].

On the other hand, the a* and b* axes, as well as the Chroma and hue components of meat coloration, are mainly influenced by the heme pigments present in meat. Myoglobin represents 90% of these, while hemoglobin and cytochromes represent the remaining 10% [43,46]. However, it has also been reported that the incorporation of plant extracts into meat and meat products can have an impact on coloration [49]. In this experiment, the addition of soursop leaf extract influenced the a*, b*, C*, and h* values.

Regarding the a* value in this experiment, differences between treatments were only observed on day 0 of chilled storage, with a lower value in the test patties supplemented with soursop leaf extract. This could be attributed to the transfer of pigments from the soursop tree leaves (mainly chlorophylls) to the patties, which decreased the a* value and increased the b* value of the patties. This phenomenon in the initial redness of the patties was also reported when avocado peel extracts were added in the preparation of pork patties, showing lower a* and higher b* values in the test patties compared to the control group patties [50]. Another factor that could be involved in the a* values in this experiment is the redox state of myoglobin. Deoxymyoglobin and oxymyoglobin keep the heme group in a ferrous state, displaying a purplish-red and bright cherry-red color, respectively. In contrast, metmyoglobin exhibits a brown color and is formed by the oxidation of the ferrous forms, transforming the heme group into a ferric state [50,51]. Reports indicate that the a* value of meat is related to the amount of oxymyoglobin [52], although if the a* values fall within a range of 4.6 to 10.8, the meat product may have a brown appearance associated with the amount of metmyoglobin [53]. The results obtained in this experiment show that the a* values of the group of patties supplemented with soursop leaf extract were not within the brown appearance perception range throughout the experimental period (4.6–10.8), while the a* value on day 10 of chilled storage for the control group patties did fall within the a* value range that could give the perception of a brown appearance associated with greater oxidation of the heme group of hemoglobin.

Regarding the b* values, they were significantly higher in the patties supplemented with soursop leaf extract on all sampling days. These results could be attributed to the pigments and phenolic compounds that have been reported to be present in soursop leaf extracts, mainly some types of flavonoids, as many of these compounds are known to act as yellow pigments [16,33,34,54,55].

In the case of C* values, differences between treatments were observed on days 5 and 10 of storage, with higher values in the group of patties supplemented with soursop leaf extract. These higher C* values indicate a greater intensity of red color in the patties [51]. There are reports that, similar to the a* value, the C* value of meat is also related to the amount of oxymyoglobin [52]; therefore, the C* data suggest that the addition of soursop leaf extract maintained the heme group of myoglobin in a ferrous state with the ability to bind oxygen to form oxymyoglobin during the 10 days of refrigerated storage. This antioxidant protection of myoglobin could be attributed to flavonoids that have been reported to be present in soursop leaf extracts from Nayarit, Mexico. Based on this, it has been demonstrated that some flavonoids such as kaempferol, myricetin, and quercetin in excess molar quantities reduce metmyoglobin, causing the heme group of myoglobin to remain in a ferrous state with the ability to bind oxygen to form oxymyoglobin [56].

The h* values showed differences between treatments on days 0 and 5 of storage, with higher values in the test patties supplemented with soursop leaf extract. These higher hue values indicate that the patties are less red due to the yellow coloration caused by the pigments added in the soursop leaf extract [51].

To evaluate the total color difference in the meat over time during refrigerated storage, the color difference equation (ΔE*) was used [57]. According to the ΔE0-10 values (Figure 2), the inclusion of soursop leaf extract in the patties had a positive effect on the color stability of the patties, as the test group showed significantly lower ΔE0-10 values compared to the control group patties. This trend in color stability for both treatments began to form during the period between 5 and 10 days of chilled storage, where differences between treatments were observed. The calculation of the ΔE* value is important, as this value allows predicting whether the observer (consumer) will notice a change in the meat color. Reports indicate that when the ΔE* value is greater than 1, the human eye can discern color differences [45,58]. Despite the lower ΔE* values in the patties supplemented with soursop leaf extract, it is likely that the consumer could detect color differences over the storage time for each of the treatments.

### 4.4. Lipid and Protein Oxidation in Mexican Hairless Pork Patties

Lipids are the nutritional component of food most prone to oxidation. In general, the oxidation of meat lipids involves a free radical chain reaction, which generates a wide range of oxidation products, such as cholesterol oxides, malondialdehyde (MDA), and 4-hydroxynonenal [48,59]. MDA is the primary secondary product of lipid oxidation, and its concentration in meat has commonly been used as an index of lipid oxidation [10,59]. In this study, the group of patties with added soursop leaf extract showed a significantly lower concentration of MDA throughout the entire experimental storage period compared to the MDA values of the control group patties. This greater stability in the lipid oxidation of patties with added soursop leaf extract is expected, as it has been proven that the addition of plant extracts can prevent the formation of free radicals and slow down oxidation; an example of this is the addition of nettle extract to ground beef, which led to a reduction in TBARS levels [60].

Currently, there is no legislative limit on the concentration of MDA in meat samples; however, various authors have established that when a food reaches values greater than 1.0 mg of MDA/kg, consumers may perceive oxidation and detect a loss of sensory quality, making the product unacceptable [57]. Other authors have also established that a concentration of 0.6 mg of MDA/kg allows for the detection of rancidity [61]. Based on the less restrictive limit of MDA concentration (1.0 mg/kg), the control group patties exceeded the restrictive rancidity threshold on day 10 of chilled storage, while the test patties with added soursop leaf extract did not exceed the rancidity threshold during the entire experimental period. Considering the more restrictive MDA concentration limit (0.6 mg/kg), the perception of rancidity in both the control group patties and the test group patties would be reflected on day 5 of chilled storage.

Regarding protein oxidation, it is characterized by causing severe modifications to the chemical structure of proteins, such as aggregation through cross-links or fragmentation through peptide cleavage [62]. These changes in the structure of proteins could affect the nutritional value and sensory quality of meat and meat products [57]. The amino acids in the side chains of proteins are the ones that undergo most oxidative modifications, with the formation of thiol groups, aromatic hydroxylation, and carbonyl group formation being the main changes [62]. Due to the changes in protein amino acids, the determination of carbonyl content in proteins has been widely used as an indicator of oxidative damage in meat proteins [10]. In this study, significant differences were observed between the patties groups on days 0 and 5 of chilled storage, with a higher amount of carbonyls observed in the patties with added soursop leaf extract on day 0 and a higher amount of carbonyls in the control group patties on day 5 (Figure 4). This higher initial concentration of total carbonyls observed in the patties with added soursop leaf extract could be influenced by the formation of stable complexes between DNPH molecules and certain flavonoids that may be present in the soursop leaf extract, as it has been reported that some types of flavonoids with carbonyl groups in their structure, such as flavanones and flavanonols, form stable complexes with DNPH [63,64].

During the period between day 0 and day 5 of chilled storage, the control group patties, lacking antioxidant protection, experienced a considerable increase in total carbonyl content, while the test group patties showed a significant decrease in total carbonyl content, likely due to the bioactive compounds added in the soursop leaf extract acting as antioxidants. In the case of flavonoids, which probably overestimated the carbonylation of meat proteins on day 0, acting as antioxidants, their structure may undergo modifications, becoming a relatively stable flavonoid radical that hardly forms stable complexes with the DNPH molecule [65]. Thus, the patties with added soursop leaf extract went from having a higher concentration of total carbonyls on day 0 to having a lower amount of total carbonyls than the control group patties on day 5. By the tenth day of storage, there were no differences in the degree of protein carbonylation between the two treatments.

The oxidative stability of lipids throughout the entire experimental period and the protection against protein oxidation until day 5 of storage in the test patties could be attributed to the bioactive compounds present in the soursop leaf extract. Although the presence and activity of carotenoids and tocopherols cannot be ruled out, flavonoids are the main group of bioactive compounds with antioxidant activity that have been reported in soursop leaves collected in the state of Nayarit, Mexico [16,29,33,34]. The reported antioxidant mechanisms of action for flavonoids that may have been involved in the patties with added soursop leaf extract include the following: (1) suppression of the formation of reactive oxygen and nitrogen species, either by inhibiting enzymes involved in their formation or by chelating free metal ions involved in free radical generation; (2) radical scavenging by donating a hydrogen atom and an electron to stabilize the radicals and form a relatively stable flavonoid radical; and (3) upregulation or protection of the antioxidant defense system [65,66,67,68].

### 4.5. Pearson Correlations Between Instrumental Color and Lipid and Protein Oxidation in Mexican Hairless Pork Patties

Among the variables in the instrumental color evaluation, lightness (L*) and the hue angle (h*) show a moderate positive correlation. This correlation is influenced by the higher lightness and the greater increase in h* value during the experimental period in the control group patties. Additionally, the a* value, b* value, and Chroma (C*) value are positively correlated with each other with moderate intensity. This correlation is influenced by the addition of soursop leaf extract to the patties, as the addition of the extract increases the b* value due to the different pigments it contains and also reduces the oxidation of hemoproteins, causing the meat to have higher a* and C* values, influenced by the oxygenation of hemoproteins [53].

Lipid oxidation is related to the discoloration of meat because there are reports indicating that lipid oxidation products can promote the oxidation of hemoproteins [42,69]. In our study, the color variables were correlated with lipid oxidation in different ways; the L* and h* values, which are influenced by the control group patties, were positively correlated with the amount of TBARS, while the a*, b*, and C* values, influenced by the patties with added soursop leaf extract, were negatively correlated with the amount of TBARS.

On the other hand, there is controversial information regarding the interaction between lipid oxidation and protein oxidation in meat. Several studies indicate that there is a correlation between these two oxidative phenomena; however, a significant number of studies report that both oxidative processes occur independently [59]. In our experiment, protein oxidation showed a weak positive correlation with lipid oxidation, suggesting that each oxidative phenomenon manifested independently. The correlations between protein oxidation and the color variables were also of low intensity.

## 5. Conclusions

The findings of this research demonstrate that the hydroalcoholic extract of soursop leaves can represent a safe and harmless source of bioactive compounds with antioxidant activity, which can be used at a concentration of 20 g leaves/kg patty as an additive in meat and meat products to reduce lipid oxidation and color deterioration after 10 days of chilled storage.

Further studies are also needed to expand our knowledge about the use of soursop leaf extracts (*Annona muricata*) in food products. Our study should be considered as a preliminary investigation that demonstrates its potential antioxidant and color-stabilizing properties in meat. We hope that these results will encourage further research to fully characterize the dose–response relationship and to define the lowest effective concentration, validate its application in different food matrices under various processing conditions and storage times, and identify and characterize specific compounds present in soursop leaves.

## Figures and Tables

**Figure 1 foods-14-03212-f001:**
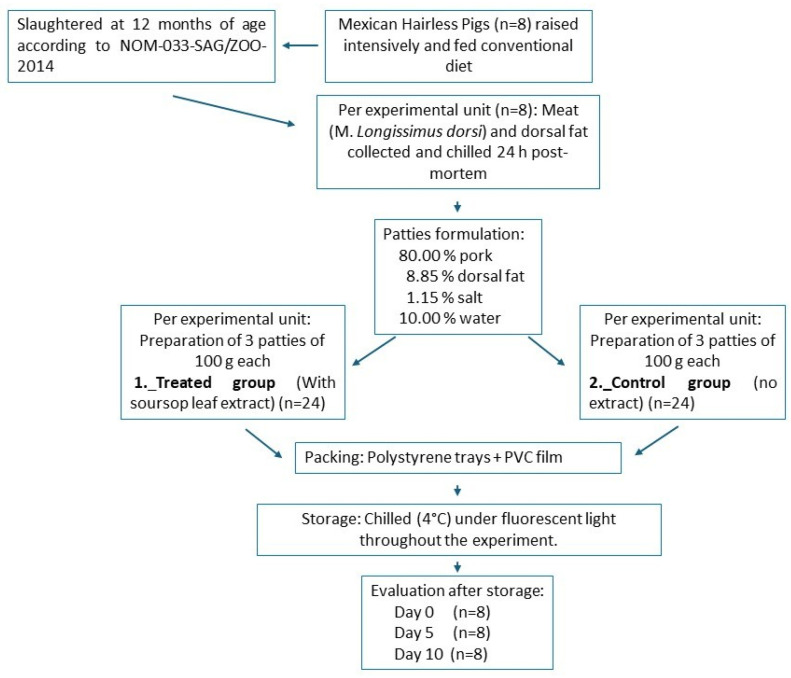
Diagram of the experimental design, patty making, and evaluation period.

**Figure 2 foods-14-03212-f002:**
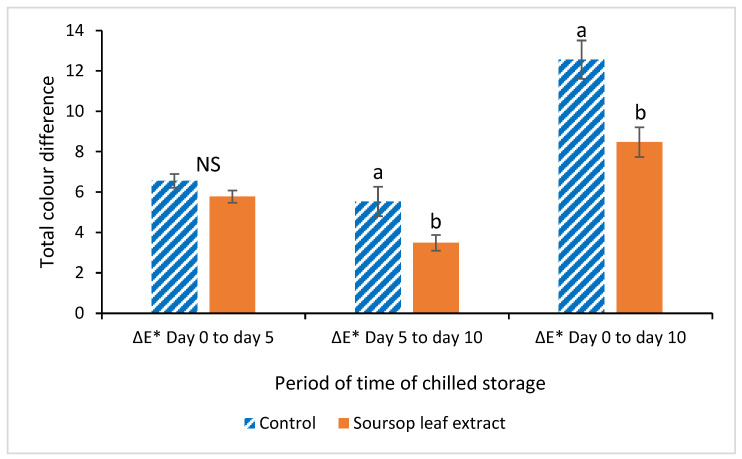
Total color difference (ΔE*) of Mexican hairless pork patties of the control group and the test group (Soursop leaf extract) between days 0 and 5, 5 and 10 and 0 and 10 of chilled storage. (^a,b^), different letters indicate differences between each treatment (*p* ≤ 0.05).

**Figure 3 foods-14-03212-f003:**
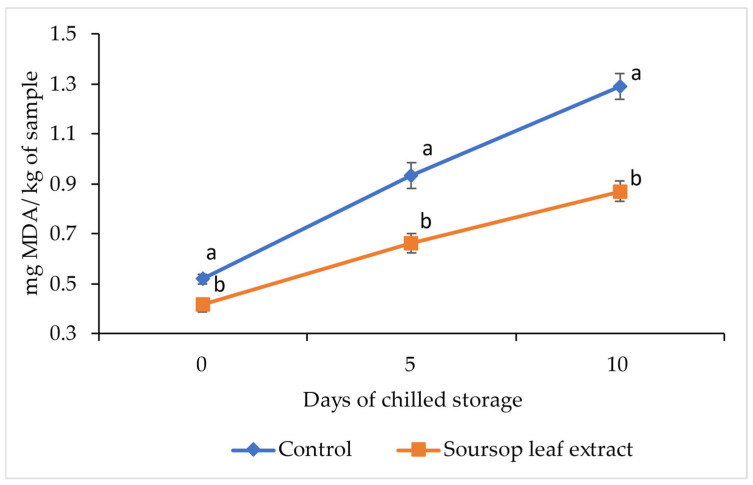
Effect of soursop (*Annona muricata* L.) leaf extract on the Thiobarbituric acid reactive substances of Mexican Hairless pork patties in chilled storage. (^a,b^), different lowercase letters indicate differences between treatments (*p* ≤ 0.05) (n = 8).

**Figure 4 foods-14-03212-f004:**
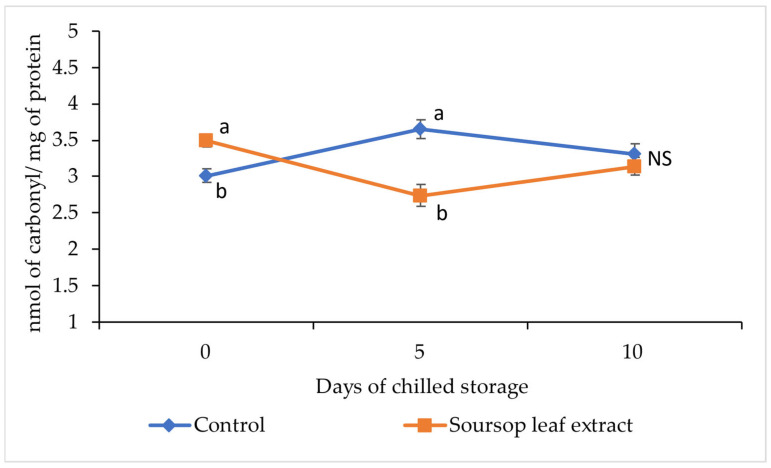
Effect of soursop (Annona muricata L.) leaf extract on protein carbonylation of Mexican hairless pork patties in refrigerated storage. (^a,b^), different lowercase letters indicate differences between treatments (*p* ≤ 0.05) (n = 8). NS, no significance.

**Table 1 foods-14-03212-t001:** Total phenolic compounds and in vitro antioxidant activity of aqueous and hydroalcoholic soursop (*Annona muricata* L.) leaf extracts.

Extraction Solvent	TPC ^A^	ABTS ^B^	DPPH ^B^
Water	202.4850	1.0264	1.1955
Ethanol-Water (70:30 *v/v*)	242.6800	1.6616	2.4418
*p*-value ^a^	0.0001 *	0.0015 *	0.0001 *
SEM	5.0071	0.1070	0.1301

TPC, Total phenolic compounds; ^A^, expressed as mg GAE/100 g fresh matter; ABTS, assay using the radical 2,2′-Azino-bis (3-ethylbenzothiazoline-6-sulfonic acid); ^B^, expressed as mmol Trolox/g fresh matter; DPPH, assay using the radical 1,1-diphenyl-2-picrylhydrazyl; SEM, standard error of the mean; ^a^, significance level in student “*t*-test”; *, significant differences (*p* ≤ 0.05).

**Table 2 foods-14-03212-t002:** Pearson’s correlation coefficient (r) between total phenolic compounds and in vitro antioxidant activity assays of aqueous and hydroalcoholic soursop (*Annona muricata* L.) leaf extracts.

	TPC	ABTS	DPPH
TPC	1	0.2827	0.6305
*p*-value ^a^		0.1616	**0.0006**
ABTS		1	0.6066
*p*-value ^a^			**0.0017**
DPPH			1
*p*-value ^a^			

TPC, Total phenolic compounds; ABTS, antioxidant activity assay using the radical 2,2′-Azino-bis (3-ethylbenzothiazoline-6-sulfonic acid); DPPH, antioxidant activity assay using the radical 1,1-diphenyl-2-picrylhydrazyl; ^a^, significance level (*p* ≤ 0.05). Bold *p*-values, significant differences.

**Table 3 foods-14-03212-t003:** Body weights ^A^ in Wistar rats treated with soursop (*Annona muricata* L.*)* leaves extract and vehicle during acute oral toxicity study.

Applied Extract	Day 0	Day 7	Day 14
Soursop leaf extract	426.40	465.56	462.80
Vehicle	456.00	464.68	489.17
*p*-value ^a^	0.31	0.97	0.47
SEM	13.89	15.16	17.26

^A^, Expressed as g; SEM, standard error of the mean; ^a^, significance level in student “*t*-test”; (n = 6).

**Table 4 foods-14-03212-t004:** Effect of soursop (*Annona muricata* L.) leaf extract on the stability of the instrumental color of Mexican Hairless pork patties in chilled storage.

	Days of Chilled Storage	Type of Pork Patties		*p*-Value ^a^	SEM
		Control	Soursop Leaf Extract		
L*	0	52.34	50.212	**0.0005**	0.3190
	5	55.6687	53.1656	**0.0014**	0.4168
	10	57.0787	53.3487	**0.0107**	0.7793
a*	0	20.5091	18.0937	**0.0001**	0.3169
	5	15.2207	14.545	0.3185	0.3321
	10	9.0087	11.4575	0.1406	0.8206
b*	0	12.3782	16.5217	**0.0001**	0.3360
	5	11.4412	14.0775	**0.0001**	0.2622
	10	11.69	13.4475	**0.0001**	0.2701
C*	0	24.0086	24.778	0.0671	0.2104
	5	18.3956	20.28	**0.0102**	0.3779
	10	15.0575	17.75	**0.0115**	0.5666
h*	0	31.3145	41.979	**0.0001**	0.8374
	5	36.9742	44.2081	**0.0001**	0.8975
	10	53.7487	49.9787	0.448	2.3830

L*, lightness; a*, red–green coordinates; b*, yellow–blue coordinates; C*, Chroma; h*, hue; ^a^, significance level in student “*t*-test”; SEM, standard error of the mean between treatments; bold *p*-values, significant differences between treatments per day (*p* ≤ 0.05) (n = 6).

**Table 5 foods-14-03212-t005:** Pearson’s correlation coefficient (r) between instrumental color, lipid oxidation and protein carbonylation of Mexican Hairless pork patties added whit soursop leaf extract in chilled storage.

	L*	a*	b*	C*	h*	TBARS	TPC
L*	1	−0.5233	−0.4969	−0.6649	0.3223	0.6138	0.1051
*p*-value ^a^		**0.0001**	**0.0001**	**0.0001**	**0.0018**	**0.0001**	0.3533
a*		1	0.2434	0.8973	−0.8618	−0.5644	−0.0095
*p*-value ^a^			**0.0239**	**0.0001**	**0.0001**	**0.0001**	0.9338
b*			1	0.4971	0.1964	−0.5215	−0.2001
*p*-value ^a^				**0.0001**	0.0699	**0.0001**	0.0895
C*				1	−0.6521	−0.7294	−0.0256
*p*-value ^a^					**0.0001**	**0.0001**	0.8223
h*					1	0.3300	−0.089
*p*-value ^a^						**0.0038**	0.4472
TBARS						1	0.1995
*p*-value ^a^							0.0513
TPC							1
*p*-value ^a^							

L*, lightness; a*, red–green coordinates; b*, yellow–blue coordinates; C*, Chroma; h*, hue; WI, whiteness index; RI, redness index; YI, yellowness index; TBARS, Thiobarbituric acid reactive substances; total protein carbonyls; ^a^, significance level (*p* ≤ 0.05). Bold *p*-values, denote significant differences.

## Data Availability

The original contributions presented in this study are included in the article. Further inquiries can be directed to the corresponding author.

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
