# Peer review of "Antioxidant Activity and Acute Oral Toxicity of Soursop (Annona muricata L.) Leaf and Its Effect on the Oxidative Stability of Mexican Hairless Pork Patties"

_foods, 2025, doi:10.3390/foods14183212_

Round 1

Reviewer 1 Report

Comments and Suggestions for Authors

Antioxidant Activity and Acute Oral Toxicity of Soursop (Annona muricata L.) Leaf and Its Effect on the Oxidative Stability of Mexican Hairless Pork Patties

This study explored the potential of hydroalcoholic extracts from soursop (Annona muricata L.) leaves as a natural antioxidant additive in meat products. The researchers examined the extract's antioxidant activity and confirmed its safety regarding acute oral toxicity. They then applied the extract to chilled Mexican Hairless pork patties to evaluate its effects on product quality. The findings showed that the soursop leaf extract was effective in reducing color loss and limiting lipid oxidation. The study concludes that soursop leaf extracts are a promising and safe source of bioactive compounds that can enhance the oxidative stability of meat and meat products.

Authors are suggested to address the following points to increase better understanding;

  1. The introduction section looks appropriate, but the authors mentioned many old references and are suggested to update the introduction with references from the last 5 to 6 years.
  2. Line 106: “and distilled” line need to be revised.
  3. Line 118: “ homogenized with 15 mL of the solvent using” For how much time the extract kept in solvent? As per most of the standard methodologies, it should be mixed at least for 3 to 6 hours for better extraction of active ingredients.
  4. Line 143: “2.4.1. Extraction of Dried Leaf Extracts for Toxicity Testing. Why is there a need to have different extraction methodologies? Does it prepare the same extract as the authors added to the products?
  5. Line 167: “using three animals.” What are the standard recommendations for animal trials? Can 3 animals give concrete results? What about replication of results?
  6. Section 2.5.1. & 2.5.2. can be presented in flow chart format for better understanding.
  7. Table 1: Instead of “TFC, Total phenolic compounds,” use TPC
  8. No need to mention A.A. in Table 1; it is already self-explanatory that ABTS and DPPH are Antioxidant activity.
  9. In Table 2. Authors changed the format as mentioned in Table 1, suggesting the use of a uniform caption or table heading related to the parameters.
  10. Justify: The small sample size of =3 per group is insufficient to confidently conclude the safety of the extract, especially given the observed adverse effects.
  11. The discussion of color changes over time is confusing; the text mentions both an increase and a period of stability for lightness (L∗), and a decrease for yellowness (b∗), requiring clearer, more consistent data interpretation.
  12. The lack of context for the observed color changes is a major issue; the paper needs to explain why the soursop extract caused a lower initial redness and a darker appearance.
  13. Reformat Table 4.
  14. The term "burgers" is used interchangeably with "patties," which is confusing and should be standardized throughout the text.
  15. Suggested to mention other crucial details about the extraction process, such as temperature, time, or the specific part of the leaf used.
  16. The discussion on acetogenins causing toxicity and death in mice directly contradicts the claim that the hydroalcoholic extract is safe and has a high LD50 value.
  17. The author suggested reducing the discussion section as it looks very lengthy.
  18. Conclusions: Very brief. Recommended to expand with key findings and a future, suggestive outline of work.

Author Response

 Comments 1: The introduction section looks appropriate, but the authors mentioned many old references and are suggested to update the introduction with references from the last 5 to 6 years.

Response 1: We greatly appreciate this observation. We have carefully revised the Introduction section and updated the references, ensuring that more than 85% of the citations are from the past six years.

Comments 2: Line 106: “and distilled” line need to be revised.

Response 2: Agree, we have carefully revised and improved the wording in line 106.

Comments 3: Line 118: “ homogenized with 15 mL of the solvent using” For how much time the extract kept in solvent? As per most of the standard methodologies, it should be mixed at least for 3 to 6 hours for better extraction of active ingredients.

Response 3: Thank you for this valuable comment. We would like to clarify that the plant material used for the extraction was not subjected to any maceration process. The extraction was performed directly on fresh plant material, rather than using a maceration method. The material was homogenized to expose as many compounds from the leaves as possible; this methodology was in accordance with that reported by Rodríguez-Carpena et al. (2011) (dx.doi.org/10.1021/jf1048832).

Comments 4: Line 143: “2.4.1. Extraction of Dried Leaf Extracts for Toxicity Testing. Why is there a need to have different extraction methodologies? Does it prepare the same extract as the authors added to the products?

Response 4: We appreciate your valuable comment. Two different extracts were prepared: one from fresh plant material and the other, for the toxicity test, from dried material. The need to prepare a specific extract for the toxicity test using dried leaves is based on OECD Guideline 423, which states that the dose must be expressed as the weight of the test substance per unit of animal body weight (e.g., mg/kg). For this reason, we prepared the extract in accordance with the requirements of this guideline, regardless of the concentration of plant metabolites (phenolic compounds).

Comments 5: Line 167: “using three animals.” What are the standard recommendations for animal trials? Can 3 animals give concrete results? What about replication of results? Response 5: We understand the reviewer's concern. However, the sample size was determined according to OECD guideline 423 [25], which is widely accepted for toxicity testing. The group of animals to which the extract was administered consisted of 6 individuals per treatment group. Guideline 423 stipulates that, for bioethical reasons, the procedure consists of 2 to 4 stages, and 3 animals must be used in each stage (as described in lines 166 and 167). Based on the findings reported in other manuscripts prior to starting the experiment, an initial dose of 2000 mg/kg body weight was administered in the first stage. Since no mortality was observed in this first stage, guideline 423 (Annex 2d) states that the same dose (2000 mg/kg) must be repeated with the same number of animals to ensure the reliability of the results; this was followed in this experiment. Because no mortality occurred in either stage after exposure to the soursop leaf extract in 6 experimental units, the guideline states that the LD50 of the extract can be determined (>5000 mg/kg) and classified according to the Globally Harmonized System (GHS) in category 5.

Comments 6: Section 2.5.1. & 2.5.2. can be presented in flow chart format for better understanding.

Response 6: That's a very good observation. Based on that, a flow chart was designed outlining the process for preparing and handling the patties (Figure 1).

Comments 7: Table 1: Instead of “TFC, Total phenolic compounds,” use TPC

Response 7: We sincerely appreciate the observation. We carefully reviewed Table 1 and Table 2 and replaced the terms with the suggested abbreviation.

Comments 8: No need to mention A.A. in Table 1; it is already self-explanatory that ABTS and DPPH are Antioxidant activity.

Response 8: Thank you very much. We agree with the suggestion and decided to remove the abbreviation A.A. “antioxidant activity” from Table 1.

Comments 9: In Table 2. Authors changed the format as mentioned in Table 1, suggesting the use of a uniform caption or table heading related to the parameters.

Response 9: Thank you for your comment; we have reviewed and standardized the format of the tables.

Comments 10: Justify: The small sample size of n=3 per group is insufficient to confidently conclude the safety of the extract, especially given the observed adverse effects.

Response 10: Thank you very much for the observation and for your concern regarding the reliability of the manuscript. The sample size consisted of six animals per treatment. OECD Guideline 423, which is a widely accepted standard for toxicological experiments, establishes that the procedure is composed of 2 to 4 steps using 3 animals in each step. The first step was carried out by administering an initial dose of 2000 mg/kg body weight. Since no mortality was observed in this first step, OECD Guideline 423 states that the same dose of 2000 mg/kg body weight should be repeated to provide reliability to the first step, and this was done in the present experiment. As no mortality occurred in the two steps after exposing our soursop leaf extract to 6 experimental units, the guideline establishes that the LD50 of the extract can be determined and classified according to the GHS. Regarding the report of clinical signs associated with adverse effects, the guideline establishes that any such signs observed within the following 14 days of administration must be reported; however, these do not modify the classification criteria of the extract.

Comments 11: The discussion of color changes over time is confusing; the text mentions both an increase and a period of stability for lightness (L), and a decrease for yellowness (b), requiring clearer, more consistent data interpretation.

Response 11: We appreciate your observation. We realized that, by mistake, the results across different time periods were not discussed in this manuscript. Therefore, we had decided to only explain the effects of the extract compared to the control group, and when submitting the manuscript, we mistakenly included the wrong table. We have now replaced Table 4 with one that only compares the treatment groups. Based on this, section 2.6 has also been revised.

Comments 12: The lack of context for the observed color changes is a major issue; the paper needs to explain why the soursop extract caused a lower initial redness and a darker appearance.

Response 12: Thank you very much for your valuable comment. In lines 537 to 548 of the manuscript, we explain the interpretation of the experimental results regarding the L* value of the color, and the possible reason why it is darker when the extract interacts with the soursop leaf extract. Regarding the interpretation of the experimental results for the a* value, this is explained in lines 556-557 and 561-566, where we highlight the influence that pigments present in natural plant extracts can have on the color changes of the final product, especially when using a colorimeter, which is a highly sensitive instrument for detecting such changes.

Comments 13: Reformat Table 4.

Response 13: Agree, we have carefully revised and improved the table 4.

Comments 14: The term "burgers" is used interchangeably with "patties," which is confusing and should be standardized throughout the text.

Response 14: Thank you very much for your valuable comment. We have carefully reviewed and standardized the manuscript with a single term (patties).

Comments 15: Suggested to mention other crucial details about the extraction process, such as temperature, time, or the specific part of the leaf used.

Response 15: Agree, we have carefully revised and improved the wording in line 123.

Comments 16: The discussion on acetogenins causing toxicity and death in mice directly contradicts the claim that the hydroalcoholic extract is safe and has a high LD50 value.

Response 16: Thank you for your observation. In the discussion section, we cite a study as background research, in which the toxicity of pure soursop leaf extracts, extracts enriched with flavonoids, and extracts enriched with acetogenins were evaluated in mice. The results reported indicate that when the pure soursop leaf extract, including the flavonoid-enriched extract, was used, there were no signs of toxicity or deaths (L517-519). This is consistent with the results of our experiment; when we used the pure extract, no signs of toxicity were observed. In this study, we did not quantify acetogenins, nor was it our objective to specifically evaluate the effect of that particular molecule.

Comments 17: The author suggested reducing the discussion section as it looks very lengthy.

Response 17: Thank you for your observation. However, we believe it would not be advisable to reduce the scope of the discussion, as all the information presented is essential to support the findings of this research.

Comments 18: Conclusions: Very brief. Recommended to expand with key findings and a future, suggestive outline of work.

Response 18: We appreciate your recommendation, the suggested modification was made.

Reviewer 2 Report

Comments and Suggestions for Authors

The manuscript addresses an interesting and relevant topic, namely the evaluation of antioxidant activity, acute oral toxicity, and the application of soursop leaf extracts in pork patties. The study is well-structured and has potential significance for functional meat products. However, in its current form, several critical issues in methodology, statistical analysis, data interpretation, and reporting must be resolved before the work can be considered for publication. My revision comments are outlined below.

  1. Unify the statistical framework

The current manuscript reports Student’s t-test in Table 4 (ll.413–416), while the Methods (ll.245–257) state that ANOVA with Tukey’s test was applied. This inconsistency must be resolved. Given the experimental design (two treatments × three storage times), a two-way ANOVA (treatment × day) followed by Tukey’s multiple comparison test is the most appropriate statistical framework. The analysis and lettering in Figures 1–3 and Table 4 should therefore be recomputed and updated accordingly. Please ensure that the statistical method is described consistently across Methods, Results, and table/figure captions.

  1. Annotate replication and uncertainty

At present, Figures 2–3 (ll.454–476) and Tables 4–5 (ll.491–503; 413–416) present only mean values with significance letters. The number of replicates (n) and the associated measure of variability (SEM or 95% CI) are not explicitly reported.

This omission reduces the transparency and reproducibility of the results. According to best practice and MDPI journal standards, figures and tables should display both sample size and dispersion metrics, enabling readers to evaluate the reliability of the statistical inferences.

  1. Control for multiple testing in correlations

In Table 5 (ll.491–503), numerous Pearson correlation coefficients are reported, each with a corresponding p-value. However, no adjustment for multiple hypothesis testing is mentioned. This raises the risk of inflated Type I error, especially since more than 10 correlations were tested simultaneously.

For example, the reported correlation between TBARS and TPC has p = 0.0513, which is already marginal without correction. After adjustment, it would likely become non-significant.

  • Apply FDR (Benjamini–Hochberg) or Bonferroni to Table 5 p-values; mark which correlations remain significant (491–503; note TBARS–TPC p = 0.0513).
  1. Align toxicity extract with the food extract

A critical methodological inconsistency exists: the toxicological assay used a dried-leaf 85:15 ethanol–water extract (ll.139–152), whereas the food application employed a fresh-leaf 70:30 ethanol–water extract (ll.115–125; 209–214). These extracts likely differ substantially in yield, phenolic composition, and metabolite profile.

As a result, the reported acute oral safety (OECD 423, ll.164–183; 312–320) cannot be directly extrapolated to the actual extract incorporated into pork patties. This undermines the toxicological relevance of the study.

  • Re-run OECD 423 using the fresh-leaf 70:30 extract used in patties (115–125; 209–214) or provide HPLC-DAD/MS compositional equivalence to the dried-leaf 85:15 extract used for toxicity (ll.139–152).
  1. Dose standardization

At present, the study reports only the amount of fresh leaves added to patties (20 g/kg, ll.209–214) and general antioxidant capacity results (Table 1, ll.265–283). However, there is no clear standardization of dose in terms of extract yield or phenolic content.

Without this, the results cannot be compared across studies, and the actual antioxidant dose per unit of meat remains unclear.

=>

  1. Report the extraction yield (%) for both water and ethanol–water extracts (weight of dried extract obtained ÷ weight of raw leaves × 100).
  2. Provide the phenolic concentration of the extracts (e.g., mg GAE/g extract) from the Folin–Ciocalteu results.
  3. Convert the inclusion level to a standardized metric: mg GAE/kg of patty for the 20 g leaves/kg formulation. This will allow other researchers to replicate and compare antioxidant efficacy across different studies.
  4. Update the Methods (ll.115–125; 209–214) to clarify the extraction yields and the final concentrations used in patties.
  5. Adjust the Discussion and Conclusions so that claims about efficacy are tied to standardized dose metrics rather than just leaf mass.
  6. Specify storage/packaging conditions

The current description of storage (ll.201–207) is incomplete: it only states “4 °C, continuous fluorescent lighting for 24 h, patties placed on polystyrene trays and wrapped with PVC film.”

Critical information about the storage environment and packaging barrier properties is missing. These parameters strongly influence lipid oxidation, protein oxidation, and color stability, and must be reported to ensure reproducibility and comparability across meat science studies.

  1. Clarify lighting regimen

In the Methods (ll.201–207), storage is described as under “continuous fluorescent lighting for 24 h.” This phrase is ambiguous: it could mean (a) patties were exposed to light only for the first 24 h, or (b) patties were continuously illuminated for the full 10-day storage.

In the Results and Discussion (ll.433–444; 619–700), the authors repeatedly attribute changes in color parameters and oxidation to light exposure, which implies lighting lasted throughout the 10 days, not just the first 24 h.

  1. Add a dose–response

In the Methods (ll.209–214), only a single inclusion level (20 g leaves/kg patty mixture) was tested. This prevents establishing whether the observed antioxidant effects follow a dose–response relationship, a critical element for evaluating the functional potential of plant extracts in meat systems.

Without dose–response data, it is not possible to identify the minimum effective dose, the saturation point, or potential pro-oxidant effects at higher levels.

  1. Temper the conclusion

The Conclusions (ll.792–799) currently state that the hydroalcoholic extract of soursop leaves “can represent a safe and harmless source… and can be used as an additive in meat and meat products.” This wording is too strong, given the limitations of the study.

Specifically, protein carbonyl levels showed no significant difference at day 10 (ll.466–467), and the experiment tested only one inclusion level (20 g/kg) in one meat model for 10 days. These limitations must be reflected in the conclusion.

  1. Detail the water-replacement step

In the Methods (ll.209–214), it is stated that patties were formulated with 20 g of leaves/kg mixture, the extract was concentrated, and then used to replace 10% of the added water. However, several essential details are missing:

The total solids content of the extract after evaporation is not reported.

The exact volume of extract solution used in substitution is unclear.

The resulting final moisture content of the patty mix is not specified.

These omissions make the formulation non-reproducible, since small differences in extract solids or moisture balance can alter patty texture, oxidation kinetics, and storage stability.

  1. Justify or measure the pigment claim for b*

The authors state that the increase in b* could be attributed to carotenoids and tocopherols, but also admit that such pigments “have not been reported in leaves from Nayarit, Mexico.” This is a logical inconsistency: attributing an observed effect to compounds that are not confirmed in the study material.

In summary, while the manuscript has potential scientific merit, it requires substantial revisions to ensure methodological consistency, statistical rigor, and reproducibility. The authors must address all the points listed above, particularly those related to statistical analysis, extract characterization, dose standardization, and the clarity of conclusions. I therefore recommend revision, with the expectation that the revised manuscript will provide more robust evidence and clearer reporting aligned with journal standards.

Comments on the Quality of English Language

The manuscript is generally understandable, but the English expression requires improvement for clarity and precision. Several sentences in the Methods and Discussion are ambiguous or overly speculative (e.g., attribution of b* changes to pigments not confirmed in the studied material, ll.666–670). Verb tense and article usage should be checked throughout to ensure consistency. In addition, some sections (e.g., storage conditions, extraction details) would benefit from more concise and precise wording to improve readability. A careful language edit by a professional editor or a fluent English speaker is recommended prior to resubmission.

Author Response

 Comments 1: Unify the statistical framework

The current manuscript reports Student’s t-test in Table 4 (ll.413–416), while the Methods (ll.245–257) state that ANOVA with Tukey’s test was applied. This inconsistency must be resolved. Given the experimental design (two treatments × three storage times), a two-way ANOVA (treatment × day) followed by Tukey’s multiple comparison test is the most appropriate statistical framework. The analysis and lettering in Figures 1–3 and Table 4 should therefore be recomputed and updated accordingly. Please ensure that the statistical method is described consistently across Methods, Results, and table/figure captions.

Response 1: Thank you for your observation. In an earlier version of the manuscript, we mistakenly included Table 4, even though we had decided not to present the treatment effects over time. For this reason, the results corresponding to the different storage periods were not discussed in the revised version. Our intention was to report only the comparison between the extract and the control group. However, during submission, the incorrect table and corresponding text were inadvertently included in Section 2.6. We have now replaced Table 4 with the correct version, which presents only the treatment group comparisons. Accordingly, Section 2.6 has been revised and improved.

Comments 2: Annotate replication and uncertainty

At present, Figures 2–3 (ll.454–476) and Tables 4–5 (ll.491–503; 413–416) present only mean values with significance letters. The number of replicates (n) and the associated measure of variability (SEM or 95% CI) are not explicitly reported.

This omission reduces the transparency and reproducibility of the results. According to best practice and MDPI journal standards, figures and tables should display both sample size and dispersion metrics, enabling readers to evaluate the reliability of the statistical inferences.

Response 2:  Thank you for your comment. In Figures 2 and 3, the standard error bars represent the variability (dispersion) of the data within each treatment, while the different letters indicate statistically significant differences (p < 0.05), as specified in the figures and their respective legends. We have revised the Methodology (Sections 2.5.1 and 2.6) to clarify the number of replicates used in each experiment, and the sample size (n) for each treatment has now been added to the table footnotes. Regarding Table 4, the standard error of the mean for each variable is presented in the last column. For Table 5, measures of dispersion cannot be included, as these values correspond to specific correlation coefficients. Given the type of statistical analysis performed, these coefficients range from -1 to 1 and indicate only the degree of association between variables.

Comments 3: Control for multiple testing in correlations

In Table 5 (ll.491–503), numerous Pearson correlation coefficients are reported, each with a corresponding p-value. However, no adjustment for multiple hypothesis testing is mentioned. This raises the risk of inflated Type I error, especially since more than 10 correlations were tested simultaneously.

For example, the reported correlation between TBARS and TPC has p = 0.0513, which is already marginal without correction. After adjustment, it would likely become non-significant.

Apply FDR (Benjamini–Hochberg) or Bonferroni to Table 5 p-values; mark which correlations remain significant (491–503; note TBARS–TPC p = 0.0513).

Response 3: Thank you very much for your observation. Indeed, statistical significance helps to determine whether there is a true relationship between the variables or if it is due to chance. Among the most common methods used are hypothesis tests, as mentioned in section 2.6; for these analyses, the Student's t-test was used.

Comments 4: Align toxicity extract with the food extract

A critical methodological inconsistency exists: the toxicological assay used a dried-leaf 85:15 ethanol–water extract (ll.139–152), whereas the food application employed a fresh-leaf 70:30 ethanol–water extract (ll.115–125; 209–214). These extracts likely differ substantially in yield, phenolic composition, and metabolite profile.

As a result, the reported acute oral safety (OECD 423, ll.164–183; 312–320) cannot be directly extrapolated to the actual extract incorporated into pork patties. This undermines the toxicological relevance of the study.

Re-run OECD 423 using the fresh-leaf 70:30 extract used in patties (115–125; 209–214) or provide HPLC-DAD/MS compositional equivalence to the dried-leaf 85:15 extract used for toxicity (ll.139–152).

Response 4: Thank you very much for your valuable comment. We acknowledge the difference between the extracts employed in the toxicological assay (dried-leaf, 85:15 ethanol–water) and in the food application (fresh-leaf, 70:30 ethanol–water). Our rationale for using the 85:15 dried-leaf extract in OECD 423 is that this preparation is more concentrated and therefore expected to contain higher levels of bioactive metabolites than the fresh-leaf 70:30 extract incorporated into patties. By evaluating the more concentrated extract, we aimed to assess the maximum potential toxicity of soursop leaves, in line with the 3Rs principles (replacement, refinement, and reduction) for the responsible use of laboratory animals.

Furthermore, OECD guideline 423 allows for a stepwise approach, beginning with lower doses and progressing to higher doses. In our study, even at the highest and most concentrated doses tested, no signs of acute toxicity were observed. Based on this outcome, it is reasonable to extrapolate that the less concentrated 70:30 fresh-leaf extract, containing lower metabolite levels, would likewise not pose acute oral toxicity risks under the conditions tested.

Comments 5: Dose standardization

At present, the study reports only the amount of fresh leaves added to patties (20 g/kg, ll.209–214) and general antioxidant capacity results (Table 1, ll.265–283). However, there is no clear standardization of dose in terms of extract yield or phenolic content.

Without this, the results cannot be compared across studies, and the actual antioxidant dose per unit of meat remains unclear.

1.Report the extraction yield (%) for both water and ethanol–water extracts (weight of dried extract obtained ÷ weight of raw leaves × 100).

2.Provide the phenolic concentration of the extracts (e.g., mg GAE/g extract) from the Folin–Ciocalteu results.

3.Convert the inclusion level to a standardized metric: mg GAE/kg of patty for the 20 g leaves/kg formulation. This will allow other researchers to replicate and compare antioxidant efficacy across different studies.

4.Update the Methods (ll.115–125; 209–214) to clarify the extraction yields and the final concentrations used in patties.

5.Adjust the Discussion and Conclusions so that claims about efficacy are tied to standardized dose metrics rather than just leaf mass.

Response 5: We thank the reviewer for these constructive comments. Regarding points 1 and 2, we clarify that the extraction yield was not determined, and the phenolic concentration of the extracts could not be expressed as mg GAE/g extract, since the extracts were not evaporated to dryness but remained in aqueous suspension. The extraction procedure employed, which provided the extracts later assessed for antioxidant activity and incorporated into the meat model systems, followed the methodology of Rodríguez-Carpena et al. (2011) (dx.doi.org/10.1021/jf1048832).

In response to points 3–5, the formulation of 20 g leaves/kg patty has been converted into a standardized metric (mg GAE/kg patty), enabling direct comparison with other studies. This information has been added to the Methods section (L217). Furthermore, the Discussion and Conclusions have been revised to frame efficacy claims in terms of standardized dose metrics (mg GAE/kg patty) rather than solely leaf mass, thereby improving both comparability and reproducibility across studies.

We believe these revisions significantly strengthen the clarity and scientific rigor of the manuscript.

Comments 6: Specify storage/packaging conditions

 The current description of storage (ll.201–207) is incomplete: it only states “4 °C, continuous fluorescent lighting for 24 h, patties placed on polystyrene trays and wrapped with PVC film.”

Critical information about the storage environment and packaging barrier properties is missing. These parameters strongly influence lipid oxidation, protein oxidation, and color stability, and must be reported to ensure reproducibility and comparability across meat science studies.

Response 6: Thank you for your observation. We have made the necessary corrections to the manuscript, including clarification that refrigerated storage conditions were maintained throughout the entire experimental period (line 209). In addition, details regarding the type of film used and the lighting conditions have been added (lines 205–207).

Comments 7: Clarify lighting regimen

In the Methods (ll.201–207), storage is described as under “continuous fluorescent lighting for 24 h.” This phrase is ambiguous: it could mean (a) patties were exposed to light only for the first 24 h, or (b) patties were continuously illuminated for the full 10-day storage.

In the Results and Discussion (ll.433–444; 619–700), the authors repeatedly attribute changes in color parameters and oxidation to light exposure, which implies lighting lasted throughout the 10 days, not just the first 24 h.

Response 7: We sincerely appreciate this observation. Indeed, the wording in the Methods section was ambiguous. We have corrected the text (L209) to specify that the patties were continuously exposed to fluorescent lighting throughout the entire storage period, and not only during the first 24 hours.

Comments 8: Add a dose–response

In the Methods (ll.209–214), only a single inclusion level (20 g leaves/kg patty mixture) was tested. This prevents establishing whether the observed antioxidant effects follow a dose–response relationship, a critical element for evaluating the functional potential of plant extracts in meat systems.

Without dose–response data, it is not possible to identify the minimum effective dose, the saturation point, or potential pro-oxidant effects at higher levels.

Response 8: We thank the reviewer for this important comment. We fully recognize that testing different inclusion levels would yield valuable insights into the presence of a dose–response relationship, the determination of a minimum effective dose, and the potential for pro-oxidant effects at higher concentrations. However, the scope of the present study was limited to evaluating a single inclusion level (20 g leaves/kg patty; 48,536 mg GAE per kg mixture), as our primary objective was to assess whether soursop leaf extract could provide antioxidant protection under practical meat product formulation conditions.

We agree that future research should explore a broader range of inclusion levels to fully characterize the dose–response relationship, identify the minimum effective concentration, and assess any potential pro-oxidant effects. This limitation has been explicitly acknowledged in the Conclusions section of the revised manuscript.

Comments 10: Temper the conclusion

The Conclusions (ll.792–799) currently state that the hydroalcoholic extract of soursop leaves “can represent a safe and harmless source… and can be used as an additive in meat and meat products.” This wording is too strong, given the limitations of the study.

Specifically, protein carbonyl levels showed no significant difference at day 10 (ll.466–467), and the experiment tested only one inclusion level (20 g/kg) in one meat model for 10 days. These limitations must be reflected in the conclusion.

Response 10: Thank you very much for this valuable comment. In response, we have revised the Conclusions to reflect the limitations of the study and the results obtained. Specifically, the revised text now acknowledges that only a single inclusion level (20 g leaves/kg patty) was tested in one meat model for 10 days. The conclusions have been tempered accordingly to avoid overstating the potential application of soursop leaf hydroalcoholic extract, while still highlighting its observed antioxidant effects under the conditions tested.

Comments 12: Detail the water-replacement step

In the Methods (ll.209–214), it is stated that patties were formulated with 20 g of leaves/kg mixture, the extract was concentrated, and then used to replace 10% of the added water. However, several essential details are missing:

The total solids content of the extract after evaporation is not reported.

The exact volume of extract solution used in substitution is unclear.

The resulting final moisture content of the patty mix is not specified.

These omissions make the formulation non-reproducible, since small differences in extract solids or moisture balance can alter patty texture, oxidation kinetics, and storage stability.

Response 12: We thank the reviewer for this important observation. In response, Section 2.5.2 of the Methods has been revised to clarify that an aqueous extract was obtained and subsequently incorporated into the patty mixture, making up the volume corresponding to the 10% water portion of the formulation. Regarding the solids content, the extract underwent double filtration during its preparation, thereby preventing the presence of solids in the final extract.

The final moisture content of the hamburger mixture was not directly measured; however, the effect of water on oxidative processes was expected to be similar in both treatments, since the aqueous extract was carefully adjusted to represent 10% of the total water portion of the mixture. Consequently, both treatments contained the same overall moisture content.

Comments 14: Justify or measure the pigment claim for b*

The authors state that the increase in b* could be attributed to carotenoids and tocopherols, but also admit that such pigments “have not been reported in leaves from Nayarit, Mexico.” This is a logical inconsistency: attributing an observed effect to compounds that are not confirmed in the study material.

Response 14: We sincerely appreciate this observation. We acknowledge the logical inconsistency in attributing the observed increase in b* values to carotenoids and tocopherols when their presence in leaves from Nayarit, Mexico, has not been confirmed. To address this, we have removed this statement from the manuscript and only attribute the observed color changes to flavonoids, which are widely reported in soursop leaves from Nayarit, Mexico.

4. Response to Comments on the Quality of English Language

Point 1: The manuscript is generally understandable, but the English expression requires improvement for clarity and precision. Several sentences in the Methods and Discussion are ambiguous or overly speculative (e.g., attribution of b* changes to pigments not confirmed in the studied material, ll.666–670). Verb tense and article usage should be checked throughout to ensure consistency. In addition, some sections (e.g., storage conditions, extraction details) would benefit from more concise and precise wording to improve readability. A careful language edit by a professional editor or a fluent English speaker is recommended prior to resubmission.

Response 1:   We thank the reviewer for this observation. We acknowledge that some sentences in the manuscript be improved for clarity and precision. the manuscript will be submitted to MDPI’s professional English language editing service to ensure it meets the highest standards of readability and scientific expression.

Round 2

Reviewer 2 Report

Comments and Suggestions for Authors

The authors have satisfactorily addressed all the concerns raised in the previous round of review. The introduction is well contextualized, methods are clearly described, and results are convincingly presented with appropriate interpretation. Figures and tables are clear, and the English is polished. No further revision is required. I recommend the manuscript in its present form.